# A generalized Flory-Stockmayer kinetic theory of connectivity percolation and rigidity percolation of cytoskeletal networks

Carlos Bueno[1,2], James Liman[1,3], Nicholas P. Schafer[1,4], Margaret S. Cheung[1,3,5,6], Peter G. Wolynes[1,4,7,8]*

**1** Center for Theoretical Biological Physics, Rice University, Houston, Texas, United States of America, **2** Systems, Synthetic, and Physical Biology, Rice University, Houston, Texas, United States of America, **3** Department of Bioengineering, Rice University, Houston, Texas, United States of America, **4** Department of Chemistry, Rice University, Houston, Texas, United States of America, **5** Department of Physics, University of Houston, Houston, Texas, United States of America, **6** Pacific Northwest National Laboratory, Seattle, Washington, United States of America, **7** Department of Physics, Rice University, Houston, Texas, United States of America, **8** Department of Biosciences, Rice University, Houston, Texas, United States of America

* pgwolynes@rice.edu

**Data Availability Statement:** The simulation trajectories and the data for the figures reported in this paper have been deposited in Zenodo (DOI: 10.

## Abstract

Actin networks are essential for living cells to move, reproduce, and sense their environments. The dynamic and rheological behavior of actin networks is modulated by actin-binding proteins such as α-actinin, Arp2/3, and myosin. There is experimental evidence that actin-binding proteins modulate the cooperation of myosin motors by connecting the actin network. In this work, we present an analytical mean field model, using the Flory-Stockmayer theory of gelation, to understand how different actin-binding proteins change the connectivity of the actin filaments as the networks are formed. We follow the kinetics of the networks and estimate the concentrations of actin-binding proteins that are needed to reach connectivity percolation as well as to reach rigidity percolation. We find that Arp2/3 increases the actomyosin connectivity in the network in a non-monotonic way. We also describe how changing the connectivity of actomyosin networks modulates the ability of motors to exert forces, leading to three possible phases of the networks with distinctive dynamical characteristics: a sol phase, a gel phase, and an active phase. Thus, changes in the concentration and activity of actin-binding proteins in cells lead to a phase transition of the actin network, allowing the cells to perform active contraction and change their rheological properties.

## Author summary

The actin cytoskeleton is a complex dynamic system, regulated by multiple proteins that bind to actin filaments. Some actin-binding proteins are crosslinkers, which can bind pairs of actin filaments, forming actin networks. Actin crosslinkers can be passive linkers, providing only structural integrity, or can be active linkers such as myosin motors, which

5281/zenodo.5645714). MEDYAN software can be found at http://medyan.org/.

**Funding:** CB, JL, NPS, MSC, and PGW were supported by the NSF Division of Chemistry RAISE grant 1743392 and by the Center for Theoretical Biological Physics, sponsored by the NSF Division of Physics grant 2019745. CB, PGW were supported by the PoLS Student Research Network sponsored by the NSF Division of Physics grant 1522550. PGW was supported by the D. R. Bullard-Welch Chair at Rice University, Grant C-0016 and by the Welch Foundation (grant C-1792). JL, MSC had additional support to use the uHPC and Sabine clusters managed by the Core facility for Advanced Computing and Data Science at the University of Houston and sponsored by the NSF Office of Advanced Cyberinfrastructure grant 1531814. The funders had no role in study design, data collection and analysis, decision to publish, or preparation of the manuscript.

**Competing interests:** The authors have declared that no competing interests exist.

exert forces on the network. Experiments have shown that crosslinked actin networks can behave viscously when the number of passive crosslinkers is low, but become elastic, when there are many crosslinkers. Motors can only lead to contraction of the network when there is an intermediate concentration of passive crosslinkers. The behavior of networks in the cell depends on the concentration and activity of several distinct crosslinkers, which have different binding sites, geometries, affinities, and concentrations. In this work we propose a simple analytical model based on chemical kinetics and the Flory-Stockmayer theory that gives us insight into how different crosslinkers interact with the actin filaments so as to give rise to the emergent mechanical behavior. This theory also allows us to compute analytically several crucial aspects of the development of the mechanical properties during network assembly.

## Introduction

Actomyosin networks are essential for crucial aspects of muscular contraction, cellular locomotion, endocytosis, the sensing of obstacles in the environment, and even for the synaptic plasticity of dendritic spines [1]. In muscle cells actin filaments and the accompanying myosin minifilaments are parallel and organized, and the mechanism of contraction is relatively well understood [2]. On the other hand, in non-muscle cells, actomyosin networks are non-equilibrium dynamic systems of actin filaments connected by actin-binding proteins [3,4]. The non-muscle actin filaments can be randomly oriented, or can form branched bundles [5,6], asters [7,8], or meshes [9,10]. The filaments can be in a homogeneous solution [5], form a distinct phase [11], or may display more complex architectures [12]. For example in neurons, the complex structure of actomyosin networks in the dendritic spines are regulated by actin-binding proteins such as non-muscle myosin IIA heavy chain (NMIIA) motors, α-actinin, actin-related protein complex 2/3 (Arp2/3), and calcium/calmodulin-dependent protein kinase II (CaMKII) [13,14].

Crosslinkers, like α-actinin, bind actin filaments at binding sites located on the sides of the filaments [15]. When the system has become sufficiently connected by α-actinin, the network rheology changes. The network behaves as an elastic solid when the concentration of α-actinin is less than the bundling threshold, but the network behaves as a viscous fluid when the α-actinin concentration is higher than the bundling threshold [5]. The stiffness of the network changes by several orders of magnitude even for small alterations of cross-linker concentration [16]. Other crosslinkers lead to a similar pattern of mechanical response depending on their structures and sizes [17]. Arp2/3 is also an actin crosslinker, but it binds and caps the minus end of a daughter filament. Thus Arp2/3 acts both as a nucleator and as a brancher [18]. Experimentally, Arp2/3 has been shown to nucleate actin filaments and form branched actomyosin networks [19]. Branched actin networks display different dynamical and rheological behavior than do randomly crosslinked networks [4,20,21]. This variety of behaviors allows actin networks with Arp2/3 to carry out distinct cellular functions. Branched actin networks can also display rare convulsive large scale remodeling events called avalanches or "cytoquakes" [21,22].

Advances in reconstituting actin myosin systems have given insight into how actin interacts with specific actin-binding proteins [23]. A reconstituted network is able to contract when the system has more than a threshold concentration of motors but only over a limited window of concentration of linkers [11]. Changing the concentration of crosslinkers in a reconstituted system with myosin affects the steady state dynamics of actin networks [24]. Experiments have

also shown that at high myosin density, crosslinkers are not needed to promote contractility on cellular length scales [25].

The variety of orientations, architectures and biochemical compositions of actin networks makes it difficult to develop a grand unifying theory that can explain all aspects of cytoskeletal contraction. Nevertheless, one of the most important factors determining the behavior of an actomyosin network is the connectivity of the network. This connectivity modulates non-monotonically the network's ability to contract [5,11,24]. Motor activity is also needed to allow actin networks to contract and modulates contractility in a non-monotonic way. Motors encourage contractility at medium levels of activity, but decrease contractility when their activity is high but their processivity is low [26–28]. The buckling of the filaments is also necessary for the contraction of highly connected actomyosin networks [29–34]. Other structural features such as filament bundling [35], the alignment of the filaments [36], branching by Arp2/3 [18], shrinking of the actin filaments [37] and global changes in the actin network architecture [38] also modulate the ability of the network to contract. All of these features depend on the biochemical composition of the actomyosin system [32].

Several models have been developed to simulate interactions between actin-binding proteins and actin filaments such as MEDYAN [39], Cytosim [40], and AFINES [41]. We have previously studied the reorganization of actin networks caused by Arp2/3 using the MEDYAN model, which includes mechanochemical feedback on the binding and unbinding of actin-binding proteins to actin filaments and represents actin filaments as mechanical objects [21]. The completeness of these simulation models is a virtue, but in this paper, we develop a simple analytical model that allows us to highlight and appreciate how the connectivity of the cytoskeletal network develops in time and influences the dynamics and rheology of actomyosin systems.

We previously have explored an actin contractility model focused on the load response of individual actin filaments and active motor-like events [31,33,42–46] and explicitly connected the concept of rigidity percolation with glass transitions in network materials [47]. In work related to the present effort, Zilman and Safran have predicted the structural behavior of non-motorized actin networks with a single crosslinker type using a theoretical mean-field model based on the Flory polymer theory [48]. These models however do not deal explicitly with the branching nature of Arp2/3, which is a key biochemical component in cortical actin. The approach taken here is based on the seminal work of Flory and Stockmayer on condensation polymer networks in a Bethe lattice[49,50]. A recent generalization of Flory's work allows us to model branched networks where multivalent monomers have multiple different binding sites, an important feature of biological actin networks [51]. We also highlight the distinction between connectivity percolation, also called conductivity percolation, which simply monitors the existence of an infinite cluster that is connected and rigidity percolation which determines when the infinite cluster becomes elastically stable [52–54]. Alvarado et al. have proposed a schematic phase diagram for active systems, with 4 regimes, where the network can be described as being an active solution, a prestressed gel, able to undergo global contractions or only local contractions [55].

In the present analysis we have developed a macroscopic chemical kinetic model based on binding and unbinding kinetics of actin-binding proteins. We found that the transient concentrations obtained from the chemical kinetic model are comparable with the results of a coarse-grained mechanochemical model (MEDYAN) before the percolation transition. We also showed how the mechanism of binding between actin-binding proteins and actin filaments and the binding cooperativity can alter the concentrations needed to observe connectivity percolation. The calculations show that low concentrations of motors are not able to produce contractile motions in the actin networks without additional linkers, but that at high

concentrations motors are sufficient to produce contractile motions. We also locate the connectivity percolation transition as a function of linker composition and explore how the rigidity percolation transition differs from the connectivity percolation transition when the connections made by the linkers are not themselves individually rigid. We find that, unlike other actin-binding proteins, Arp2/3, an actin brancher that generates complex architectures, modulates the actomyosin percolation in the network in a non-monotonic way. In conclusion, the present model based on the Flory-Stockmayer theory allows us to determine how the biochemical composition, branching, and the linker binding mechanism are linked to the connectivity in the system and the observation of contraction.

## Results

### Macroscopic chemical kinetics laws recapitulate MEDYAN simulations of binding stoichiometries

We used a macroscopic kinetic description to predict the number of both the connections made during the growth, and the number of plus and minus ends of the actin filaments, which determines the length and their treadmilling rate. We compared the transient concentrations of the different kinds of F-actin binding species obtained from a chemical kinetic model to the predicted transient concentrations of F-actin binding species obtained using MEDYAN. MEDYAN is a state-of-the-art coarse-grained mechanochemical model of the actomyosin networks. MEDYAN, unlike the chemical kinetic model, includes stochastic chemical reactions, mechanical representations and mechanochemical feedback of far-from-equilibrium systems. The chemical kinetic model allows us to find an analytical solution to the percolation of actomyosin networks over time.

In the main, the transient concentrations from the chemical kinetic model and MEDYAN simulations agree with each other as shown in Fig 1. For both models we started the system with small filaments of F-actin that act as nucleators. During the first part of the trajectories, G-actin polymerizes into F-actin filaments and actin binding sites become available for actin-binding proteins, such as α-actinin, myosin, and Arp2/3 for them to bind. As the simulation progresses the binding and unbinding rates even up and the concentrations of bound actin-binding proteins reach a steady state.

Late in the growth of the network there are some differences between the transient concentrations of bound sites predicted by the chemical kinetic model and those from the MEDYAN simulation. First, the concentration of F-actin monomer direct connections ($[F_m \cdot F_p]$) obtained from MEDYAN was slightly lower than the chemical kinetic model result (purple lines in Fig 1). We attribute this small difference to the fact that, in MEDYAN, the polymerization rate of those filaments that are near the wall is decreased by mechanochemical feedback when they collide with the wall. The chemical kinetic model does not take such mechanochemical feedback or wall interactions into account.

The concentration of bound motors ($[F_c \cdot M \cdot F_c]$) obtained from the MEDYAN simulation does not differ from the concentration of bound motors in the macroscopic chemical kinetic model (yellow lines in Fig 1). On the other hand, the concentrations of bound linkers ($[F_c \cdot L \cdot F_c]$) obtained from MEDYAN differs from the concentrations of bound linkers calculated using the chemical kinetic model (orange lines in Fig 1). This discrepancy arises from the heterogeneous distribution of the binding sites in the system. In the chemical kinetic model, a homogeneous distribution of binding sites and an isotropic network conformation is assumed, while in MEDYAN the distribution is spatially heterogeneous and the filaments can form bundles. The heterogeneous distribution of binding sites in MEDYAN implies that fewer binding sites are available at a given time to be bound by linkers due to the small distance of search that

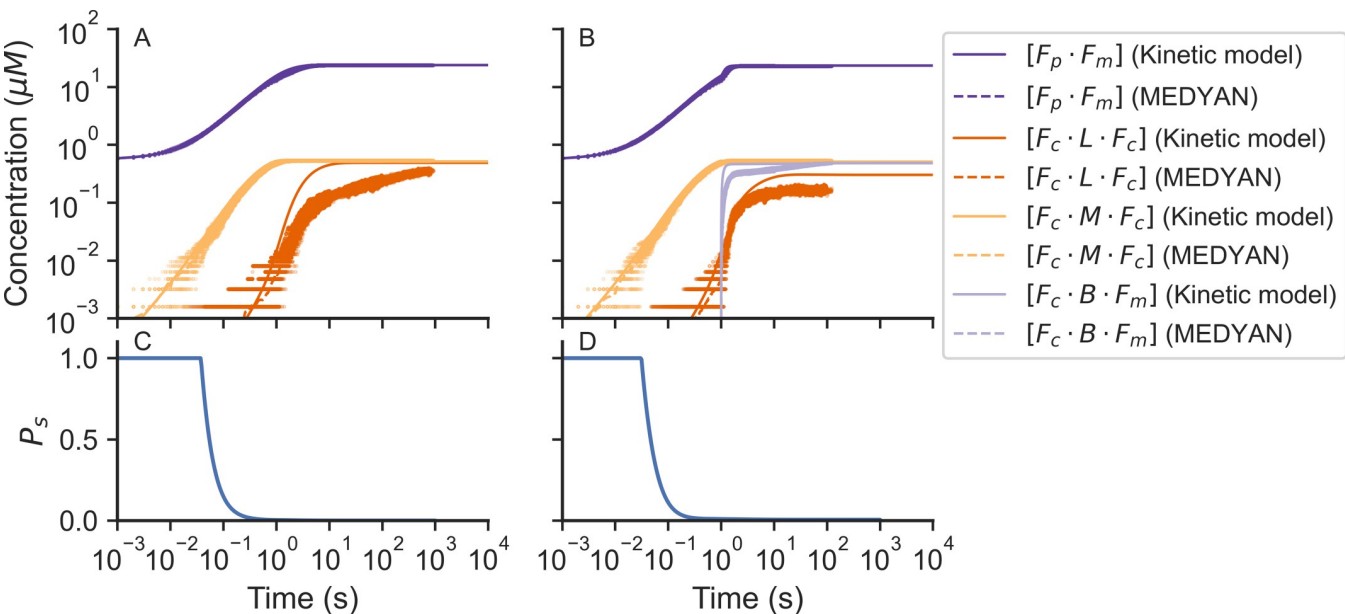

**Fig 1. The time course of the concentrations of bound species provided from the macroscopic chemical kinetic model (solid lines) and the MEDYAN simulations (dots) of actomyosin networks.** Results for networks without Arp2/3 are shown in (A) and results with Arp2/3 are shown in (B). The average MEDYAN concentrations are plotted as dotted lines. Fraction of F-actin monomers in finite clusters obtained from the chemical kinetic model are shown in (C) without Arp2/3 and results with Arp2/3 are shown in (D). $[F_m \cdot F_p]$ is the concentration of plus sites of F-actin monomers bound to a minus site of another actin monomer. $[F_c \cdot L \cdot F_c]$ is the concentration of F-actin monomer binding sites bound to another F-actin monomer binding site through a linker (α-actinin). $[F_c \cdot M \cdot F_c]$ is the concentration of actin monomer binding sites bound to another actin monomer binding site through a motor (NMIIA). $[F_c \cdot B \cdot F_m]$ is the concentration of actin monomer binding sites bound to a minus site of another actin monomer through a brancher (Arp2/3).

is possible for linkers that have already been bound ($d_C^{min} = 30$ nm, $d_C^{max} = 40$ nm). The concentration of bound motors ($[F_C \cdot M \cdot F_C]$) acquired from MEDYAN and from macroscopic kinetics are similar because the search distance for a motor is greater than the search distance for a linker in the MEDYAN model ($d_M^{min} = 175$ nm, $d_C^{max} = 225$ nm).

Finally, the concentration of bound branchers ($[F_c \cdot B \cdot F_m]$) found in MEDYAN is greater than the concentration of bound branchers obtained from the chemical kinetic model (red lines in Fig 1). This difference is a consequence of the results for other species that we have just discussed. The large concentration of non-polymerized G-actin molecules in the system with Arp2/3 (Fig 1B) predicted by the MEDYAN model comes from there being a slower effective polymerization rate caused by collisions between actin filaments and the boundary. The resulting larger concentration of available binding sites allows the branching reaction to occur faster in MEDYAN compared with what happens in the macroscopic chemical kinetic model. The concentrations at steady state of the bound species in MEDYAN tend to converge to those from the chemical kinetic model.

## Actomyosin networks undergo two sol-gel transitions when modeled using the two-step model of linker binding

Some actomyosin models simplify the binding of linkers and motors to F-actin filaments as a one-step reaction, binding two actin filaments at the same time [39,40]. Other models consider a two-step model for actin binding, where each reaction happens at different moments [5,56,57]. Here we examine the effect of a two-step non-cooperative model on the percolation of actin networks where both actin-binding domains have the same affinity to actin-binding sites.

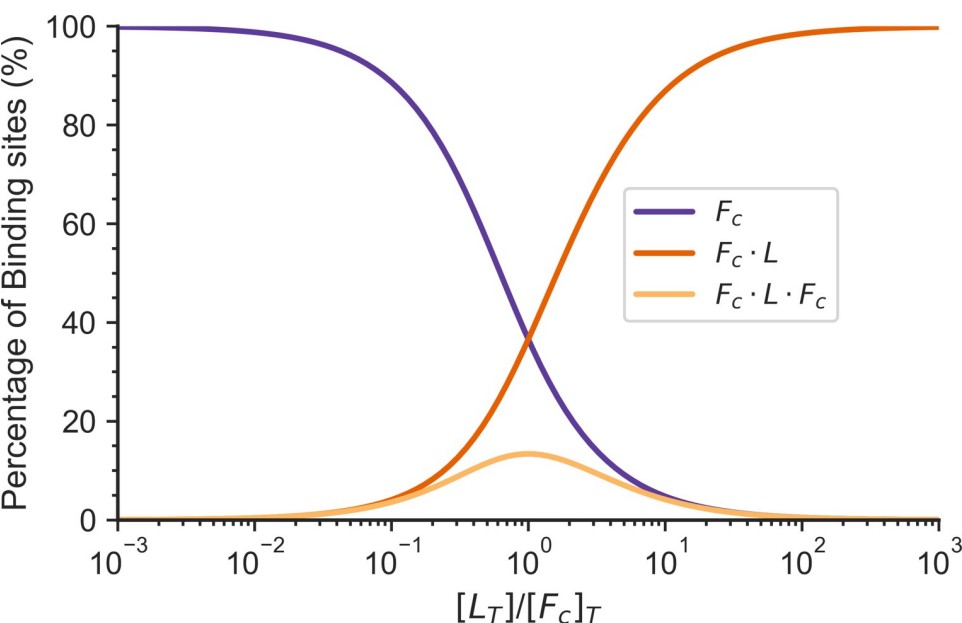

**Fig 2. The percentage of binding sites to the total number of binding sites in different states for a two-step linker binding model is shown as a function of the total number of linkers in a system.** $F_c$ is the percentage of the concentration of free binding sites to the total concentration of binding sites. $F_c \cdot L$ is the percentage of the concentration of binding sites bound to a linker to the total concentration of binding sites. $F_c \cdot L \cdot F_c$ is the percentage of the concentration of crosslinks to the total concentration of binding sites. The total concentration of binding sites $[F_c]_T$ in the system is 25 μM, $k_c^+ = 1$ μM$^{-1}$s$^{-1}$, $k_c^- = 1$s$^{-1}$.

As they grow, actomyosin networks undergo a sol-gel transition. In the two-step model of linker binding, both heads of the linker bind independently to the actin-binding sites (non-cooperative binding). At an intermediate linker concentration, a maximum concentration of crosslinker connections ($[F_c \cdot L \cdot F_c]$) is found. Above this linker concentration the binding sites have become saturated with linkers, increasing the single bound linker concentration ($[F_c \cdot L]$). The number of crosslinker connections is maximum when the concentration of linkers is equal to the concentration of binding sites, as shown in Fig 2.

For this model both heads of the linker have the same probability to bind to an actin-binding site, independent of the state of the opposite head. The number of connections in this system depends on multiple factors, including the total concentration of binding sites ($[F_c]_T$), the total concentration of linkers ($[L]_T$), and the linker binding equilibrium constant ($K_c$). The number of connections formed in the system is shown in a 2D plot by normalizing the number of connections with the total concentration of binding sites.

The maximum concentration of crosslinker connections ($[F_c \cdot L \cdot F_c]$) for this system occurs when the binding constant ($K_c$) is larger than the total concentration of binding sites ($[F_c]_T$) and the total linker concentration ($[L]_T$) is equal to the total concentration of binding sites ($[F_c]_T$) (Fig 3). A small survey of experiments in the literature shows that the first connectivity percolation transition has been observed when the system is not saturated by linkers. (Fig 4).

## Arp2/3 complex changes the percolation threshold

To understand the effects of motors and Arp2/3 complexes on the connectivity of the actomyosin network we included them in the analysis of the macroscopic chemical kinetic model and calculated the probability that an actin monomer is in a finite cluster ($P_s$). When $P_s < 1$, there is at least one infinite cluster in the system, and the system has formed a gel.

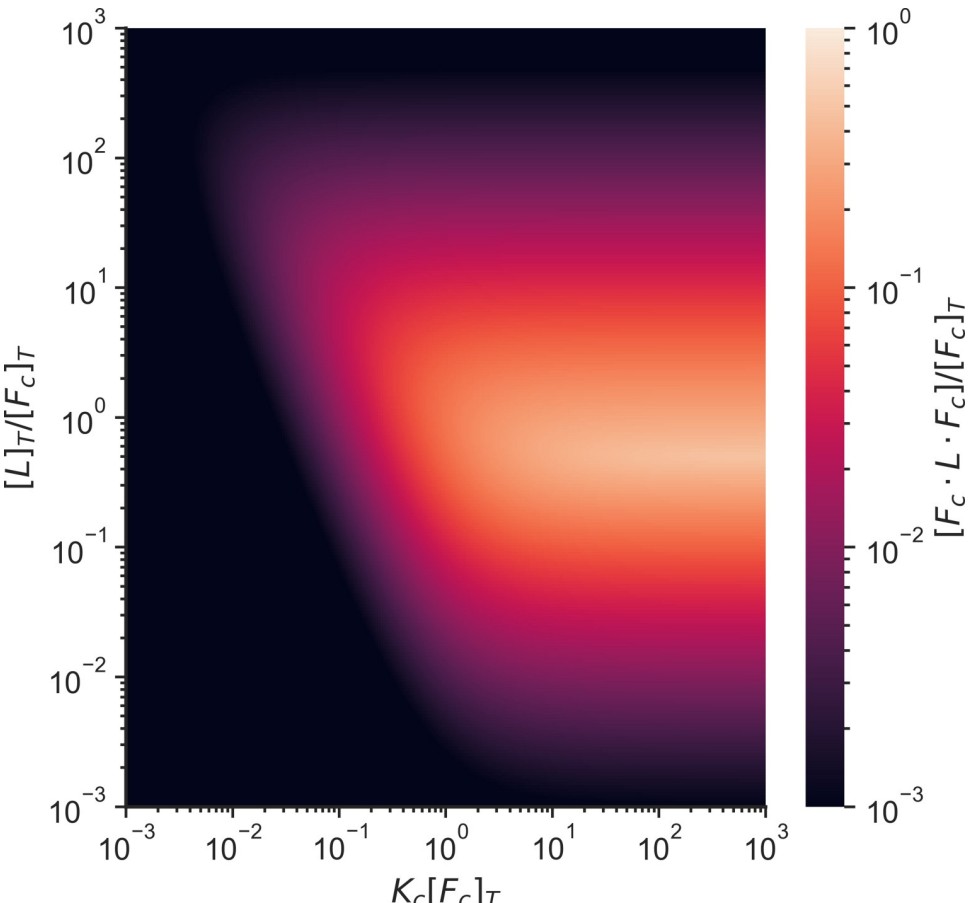

**Fig 3. Heatmap of the proportion of the concentration of crosslinks to the concentration of total binding sites as a function of the linker binding equilibrium constant and the concentration of linkers.** All axes have been normalized by the concentration of total binding sites in the system. $[L]_T$ is the total linker concentration, $[F_c]_T$ is the total concentration of binding sites, $[F_c \cdot L \cdot F_c]$ is the concentration of crosslinks, and $K_c$ is the linker binding equilibrium constant.

Motors connect the system in much the same way as linkers do since they also connect two binding sites. Therefore, motor binding increases the crosslinking probability ($p_c$) in an additive manner with linker binding (Fig 5A). Including 0.5 μM Arp2/3 to the system reduces the total number of linkers or motors required to gelate the network (Fig 5B).

Increasing the Arp2/3 concentration decreases the concentrations of linkers or motors needed to gelate the network. Only at high Arp2/3 concentrations do we find that the system is unable to form a gel even in the presence of high linker concentration (Fig 6). This is due to the saturation of binding sites by Arp2/3 which competes with linker binding, and the saturation of minus sites ($F_m$) which compete with polymerization.

## Contraction occurs when the network is gelated by motors and linkers and not gelated by only linkers

Linkers and motors behave differently in actomyosin networks. Motors tend to walk over filaments, exerting forces in the network, while linkers act as structural beams making the network more rigid. In the previous section we defined percolation as occurring when the network is simply fully connected by a combination of linkers and motors, but it is also

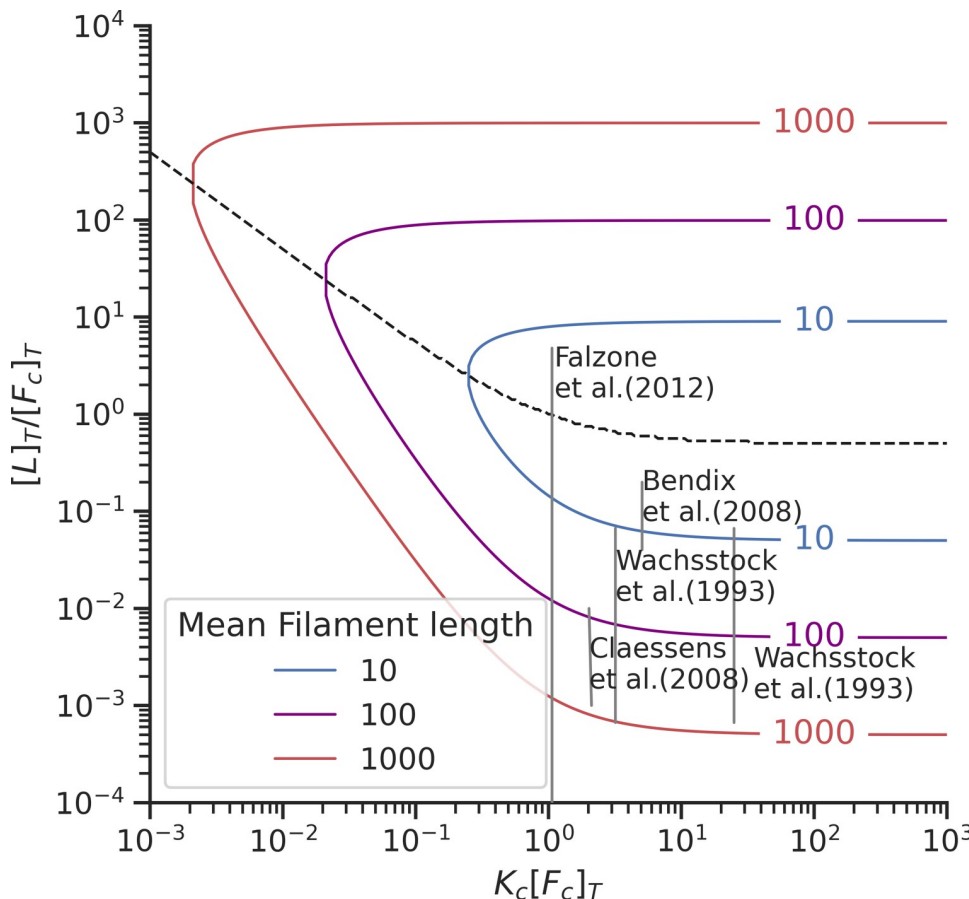

**Fig 4. Plot showing the location of different experiments on actin crosslinking plotted in the two-step model phase space [5,11,58,59].** The curved lines indicate percolation transitions for filaments of different sizes. The dotted black line indicates the region where the maximum number of crosslinks can be observed. $[L]_T$ is the total linker concentration, $[F_c]_T$ is the total concentration of binding sites, and $K_c$ is the linker binding equilibrium constant.

possible to define a connectivity percolation limit monitoring when the system is connected by linkers alone (Fig 7).

Taking into consideration the different behavior of motors and linkers, we see there are three regimes for our system (Fig 8). In a first regime at low concentrations of motors and passive linkers (purple region in Fig 8) the system cannot form a gel. In a second regime at high motor concentration and low linker concentration (white region in Fig 8) the system is gelated by motors but does not form a gel by linkers considered by themselves. Finally in a third regime at high linker concentration the system is gelated by linkers and motors acting together (green region in Fig 8).

When both the linker and motor connections with the actin are individually rigid, the number of degrees of freedom lost by binding equals the total number of degrees of freedom of the actin monomer, therefore the threshold for connectivity percolation will be the same as for rigidity percolation. Noting this, we suggest that these three regimes can explain the different mechanical behaviors manifested by the actomyosin network. In regime 1, the system is floppy and cannot transmit or exert forces. In regime 2, the motors can exert forces to the system and the system is able to contract. In regime 3, the linkers provide structure to the network so the network can transmit forces, but the network has become so rigid that it is unable to contract significantly through motor action.

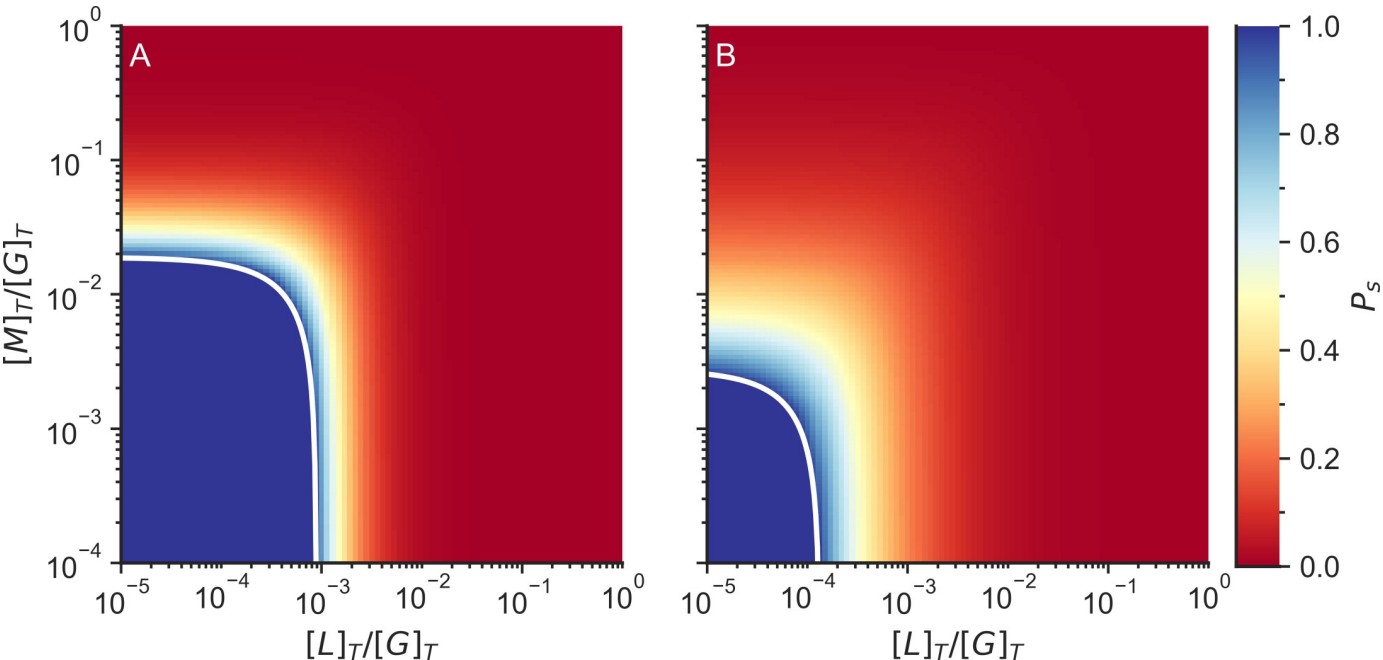

**Fig 5. Fraction of actin monomers in finite clusters ($P_s$) without Arp2/3 (left) and with Arp2/3 (right).** The color indicates the probability that an F-actin monomer is in a finite cluster. The white line indicates the connectivity percolation transition. The system is not gelated when $P_s = 1$, while the system is gelated when $P_s < 1$. $[L]_T$ is the total linker concentration, $[M]_T$ is the total motor concentration, and $[G]_T$ is the total G-actin concentration. The total concentration of G actin in the system was 25 μM and the total concentration of Arp2/3 on the simulations with Arp2/3 was 0.5 μM.

When the individual linker connections are flexible, the number of degrees of freedom of the system depends on the number and the rigidity of individual linker connections. A totally rigid connection takes away 6 degrees of freedom from the system, while a connection that only preserves the distance between two monomers would only take away one degree of freedom from the system. Fascin, a small globular crosslinker, creates rigid connections [60] that take away 6 degrees of freedom once formed, while forming a crosslink with α-actinin, takes away only 1 degree of freedom when the system is at rest, since the actin-binding domain of alpha actinin can rotate and bend with respect to the rod domain [61]. In general, as the linker connections become more flexible, more linker connections are required to reach the rigidity percolation transition (Fig 9).

## Discussion

### Arp2/3 changes the requirements for the network to percolate

There are 3 possible regimes of mechanical behavior that depend on connectivity. In the regime where neither linkers nor motors form a percolated cluster, any forces exerted by the motors cannot be transmitted through the system over large distances; the system therefore does not contract and will exhibit only local fluctuations. In the regime where the linkers alone do not form a percolation cluster, but the motors and linkers together do, the system is not yet rigid, but the forces can be transmitted throughout the system, allowing global contraction. In the regime where the linkers by themselves percolate, the system becomes highly rigid so that the motors are unable to contract the system. At high motor concentrations, motors also act as crosslinkers and can form a percolation cluster and allow by themselves contraction. This picture obtained from the Flory-Stockmayer kinetic analysis agrees with experimental results that

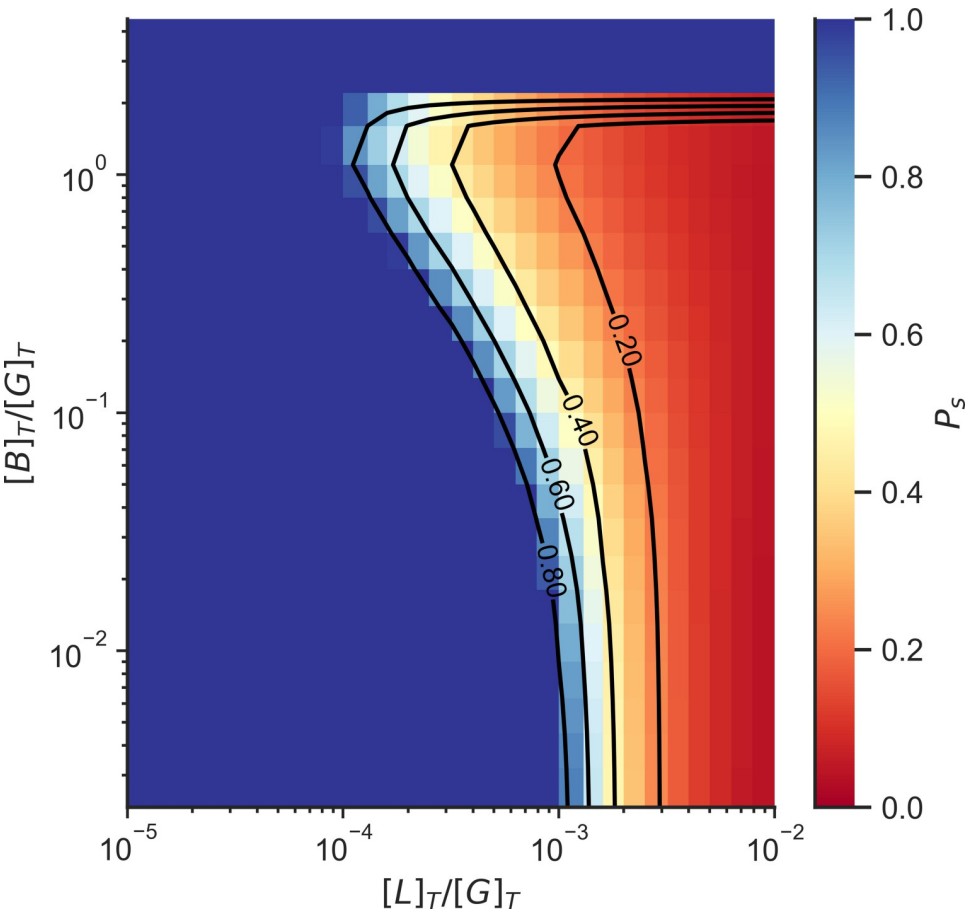

**Fig 6. Fraction of actin monomers in finite clusters ($P_s$) as a function of Arp2/3 concentration and crosslinker concentration with the exception of motors.** The color indicates the probability that an F-actin monomer is in a finite cluster. The system is not gelated when $P_s = 1$, while the system is gelated when $P_s < 1$. $[L]_T$ is the total linker concentration, $[M]_T$ is the total motor concentration, and $[G]_T$ is the total actin concentration. The total concentration of G actin in the system was 25 μM.

indicate that contraction can occur at high motor concentrations even without the presence of other crosslinkers [25].

Arp2/3 increases the connectivity and the rigidity of the network, allowing the system to exhibit global contraction at smaller concentrations of crosslinkers and motors. Arp2/3 also makes the network rigid at smaller concentrations of linkers. At high concentrations of Arp2/3, however Arp2/3 reduces the average size of the filaments, and when the concentration of Arp2/3 becomes larger than the concentration of F-actin monomers, the network becomes disconnected. The limit for rigidity percolation coincides with the limit for connectivity percolation when the individual motor and linker connections are rigid by themselves.

Each newly formed connection between F-actin monomers and an actin cluster adds a new monomer and six degrees of freedom to the cluster in the Bethe lattice percolation model. When these connections are rigid, each connection also removes six degrees of freedom from the cluster, keeping the cluster as a rigid object. In contrast when the connections are flexible so that monomers can bend or slide while remaining together, each connection removes only up to five degrees of freedom, allowing the cluster to remain flexible.

If the crosslinks formed by the linkers and the motors are rigid, the three connectivity percolation regimes shown in Fig 10 coincide with the rigidity percolation regimes. The rigidity

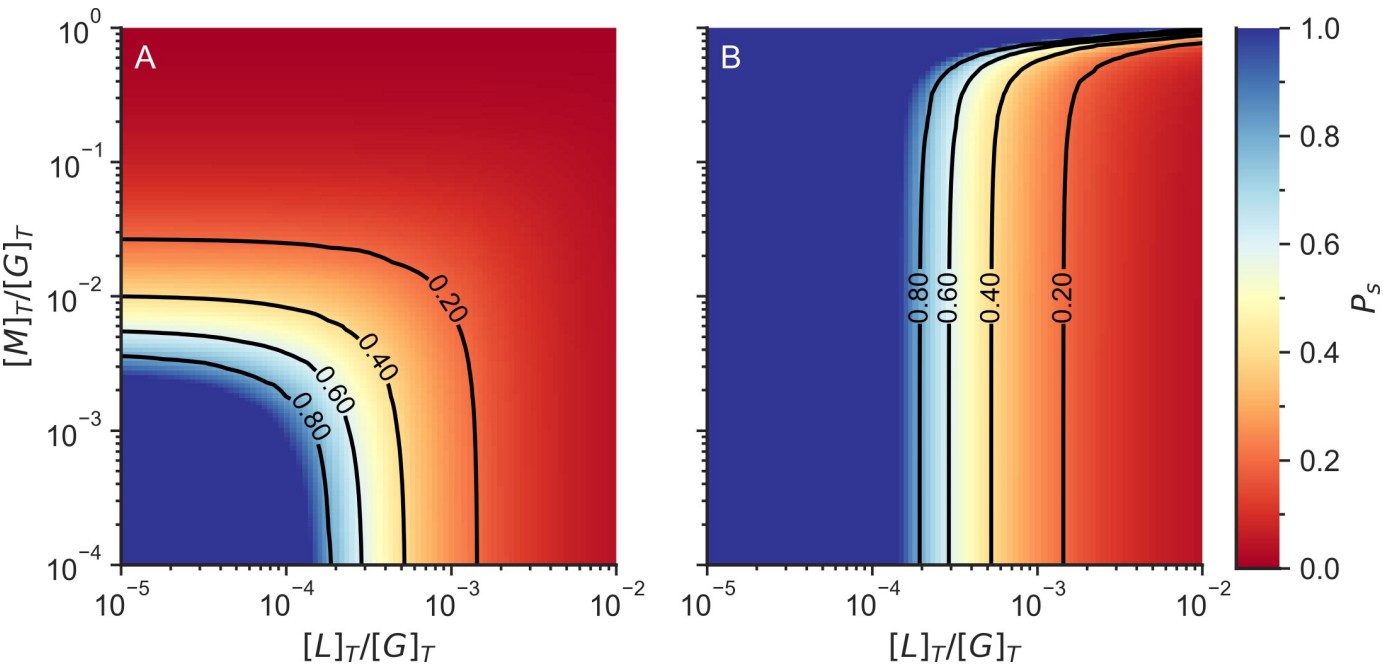

**Fig 7. Fraction of actin monomers in finite clusters ($P_s$) including motor and linker crosslinks (A) or only when considering linker crosslinks (B).** The color indicates the probability that an F-actin monomer is in a finite cluster. The system is not gelated when $P_s = 1$, while the system is gelated when $P_s < 1$. $[L]_T$ is the total linker concentration, $[M]_T$ is the total motor concentration, and $[G]_T$ is the total actin concentration.

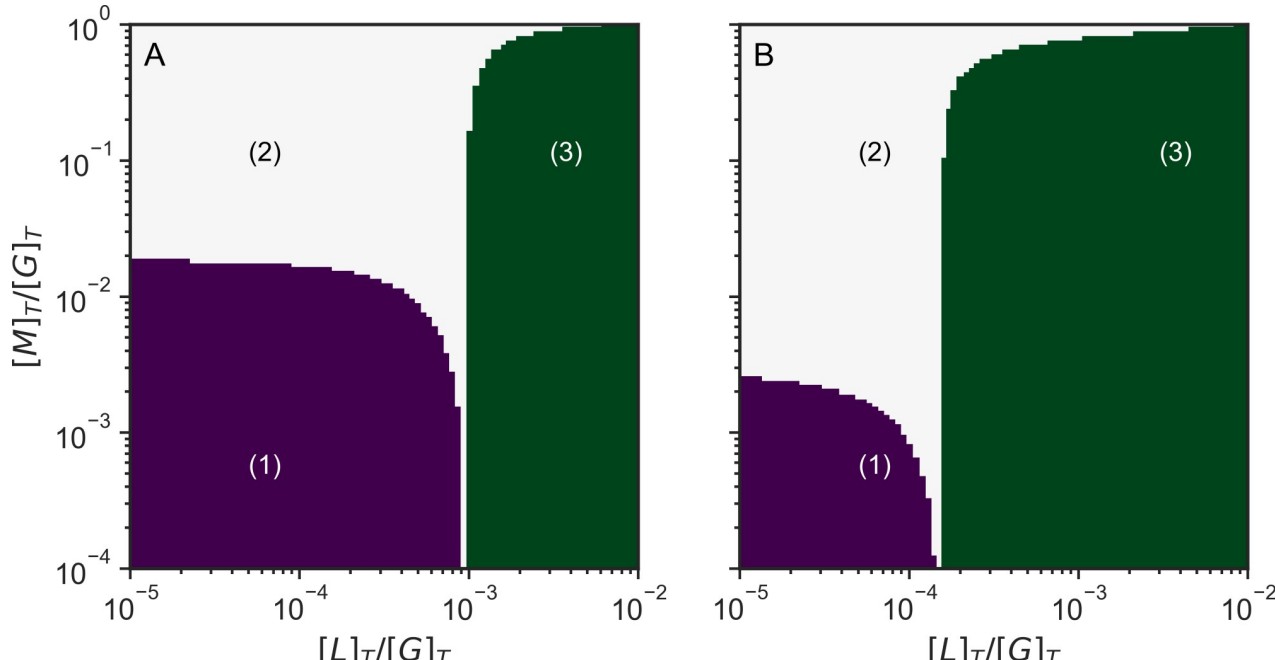

**Fig 8. Schematic phase diagrams of actomyosin systems as a function of linker and motor concentrations of actomyosin networks without Arp2/3 (A) and with Arp2/3 (B).** In region (1) the system is not gelated. In region (2) the system is not gelated only by linker connections, but the system is connected fully when we also consider the motor connections. In region (3) the system is gelated just by linkers alone. $[L]_T$ is the total linker concentration, $[M]_T$ is the total motor concentration, and $[G]_T$ is the total actin concentration.

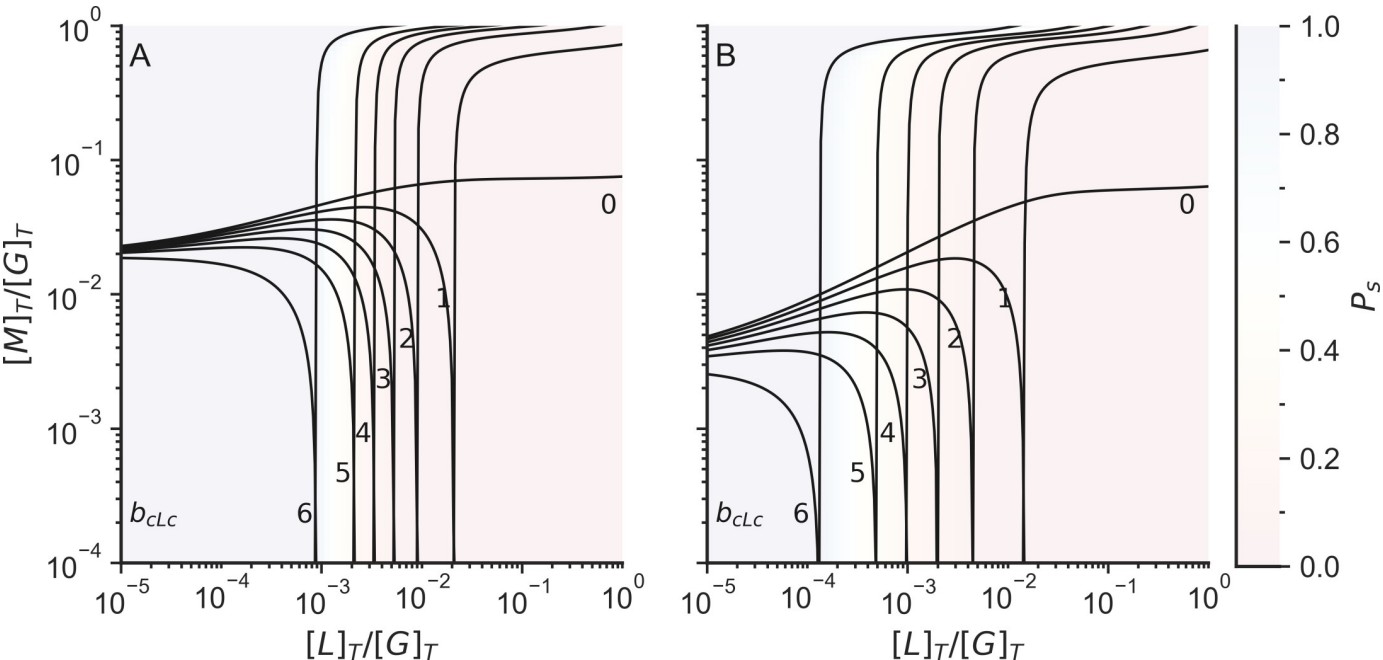

**Fig 9. Rigidity percolation limits including both motor and linker crosslinks (left lines) or those found when only considering linker crosslinks (right lines).** The number indicates the rigidity of the crosslinkers ($b_{cLc}$). We assume that the connections between plus and minus sites, the connections between binding sites and minus sites, and the connections between motors and binding sites are rigid ($b_{p \to m} = b_{c \to m} = b_{cMc} = 6$). $[L]_T$ is the total linker concentration, $[M]_T$ is the total motor concentration, and $[G]_T$ is the total actin concentration. The color of the background indicates the probability that an actin is in a finite cluster ($P_s$).

assumption is valid if the linkers and the motors do not bend or slide along filaments. The assumption of complete rigidity may be strictly valid for only some crosslinkers such as fascin and espin. Arp2/3 branchers and the connections formed between actin monomers act as rigid connections. For more flexible crosslinkers such as α-actinin and filamin, the rigidity approximation does not hold since these types of crosslinkers show high flexibility between their actin-binding domains and the rod domains [61]. Rigidity and force propagation through other mechanisms in the actin network, such as hydrodynamics, could also play a significant factor in contraction. These mechanisms are outside of the scope of this paper.

An actin system having only highly flexible linkers will not reach rigidity percolation regimes unless closed loops are formed in the system. The chemical kinetics model combined with the Flory-Stockmayer theory, is implemented on a Bethe lattice, which does not contain closed loops. Bethe lattice models can be made to account for rigid percolation regimes by anchoring multiple monomers to a single boundary, as shown in previous literature [52,62]. A complete theoretical treatment of a rigidity percolation model of actomyosin networks must deal with the formation of closed loops in such networks.

### The actin network is in the sol state when the linker concentration is much greater than the actin concentration

In our previous works [21], we modeled linker binding as a termolecular reaction in which a linker must simultaneously bind two actin filaments, forming a crosslink. Termolecular reactions in biology can however be decomposed into two separate bimolecular steps. Here we explored the behavior of a non-cooperative linker binding where the binding sites are distributed homogeneously. The binding rate constants for these actin-binding domains with actin filaments are independent of each other. Under this condition, actin-binding domains of

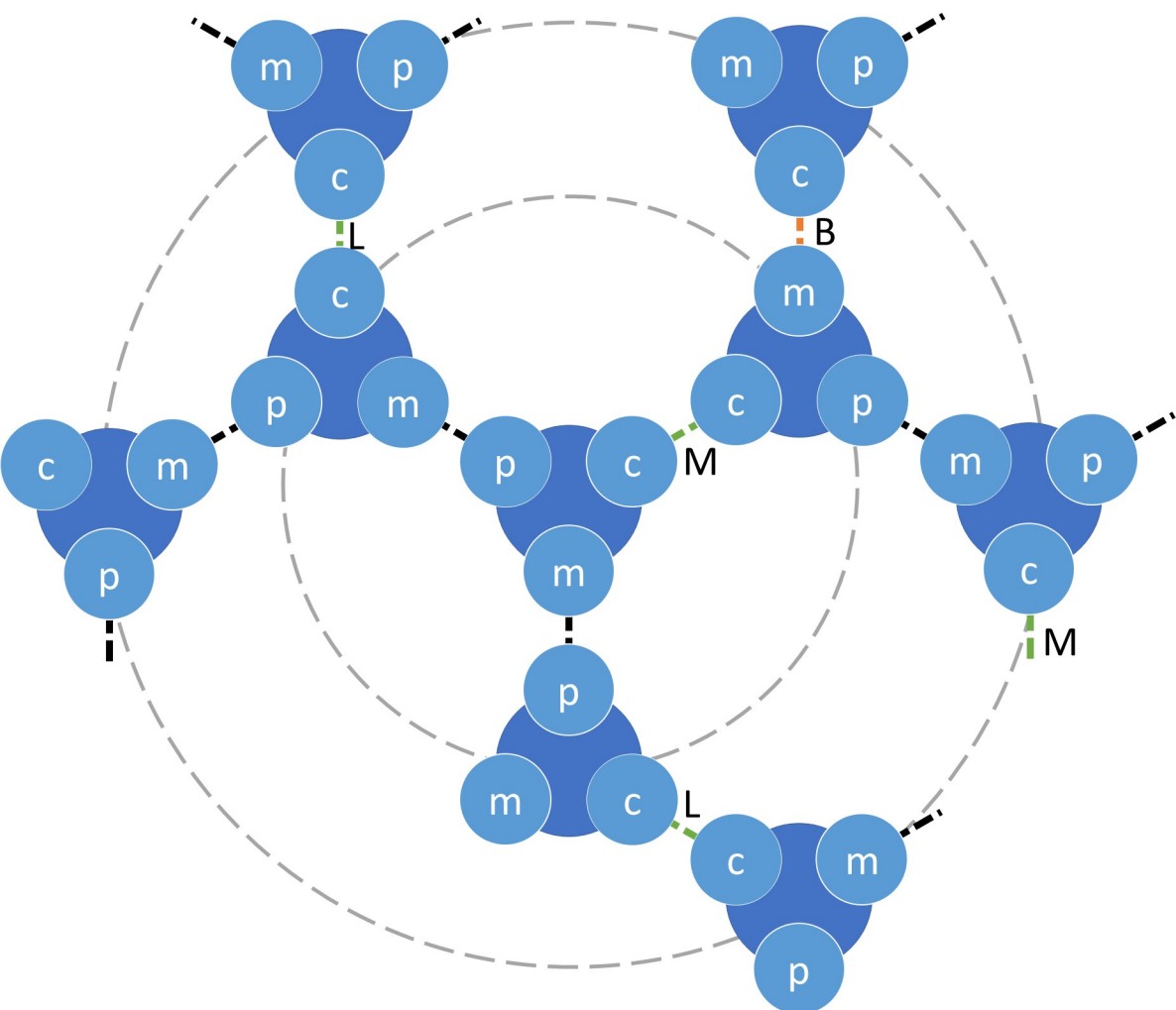

**Fig 10. Diagram of connections of F-actin monomers to other F-actin monomers.** The F-actin monomers are shown in blue and have 3 sites: the plus site (p), the minus site (m), and the binding site (c). The dotted lines indicate connections from the site of an F-actin monomer to another F-actin monomer. The connections are formed by polymerization (black dotted lines), linkers (L), motors (M) or branchers (B). The actin cluster can be represented as a treelike cluster, where the particle in the center is the root, and can be connected to up to 3 particles in the first layer, 6 particles in the second layer, and so on.

different linkers compete for filament binding, inhibiting the formation of crosslinks between two distinct filaments in the system (see Fig 2). This mechanism causes the network to remain a sol when the linker concentration is much greater (10–100 times) than the actin concentration (see Fig 4). However, previous experiments have shown that the actin-binding domains of some linkers such as α-actinin and filamin bind independently to actin filaments in a cooperative way, when actin bundles form [5]. A full theoretical treatment of cytoskeletal percolation must deal both with the bundling of actin filaments and the cooperativity of linker binding.

## The difference between the chemical kinetic model and MEDYAN can be attributed to the heterogeneous distribution when the system is percolated

Despite the results from chemical kinetic differential equations and from MEDYAN showing similar trends, there are noticeable differences of the transient concentrations of plus sites bound to minus sites ($[F_m \cdot F_p]$), bound linkers ($[F_c \cdot L \cdot F_c]$), bound branchers ($[F_c \cdot B \cdot F_m]$) in the systems

between the two models. These differences occur once the system has formed a percolation cluster. The differences for the transient concentration of plus sites bound to minus sites ($[F_m \cdot F_p]$) are caused by the diminishing polymerization rate of those filaments that are near the boundary due to the mechanochemical feedback in MEDYAN. An infinite system without boundary has been assumed in the chemical kinetic model, therefore this mechanical feedback does not occur in that model. We attribute the discrepancy in the transient concentrations of the bound linkers ($[F_c \cdot L \cdot F_c]$) to the heterogeneous distribution of the binding sites in the system in MEDYAN. In the chemical kinetic model, in contrast a homogeneous distribution of binding sites is assumed.

There are two types of structural connections formed between actin monomers: lateral connections and longitudinal connections. In this work, we have only used only the lateral connection in our model to calculate the connectivity percolation. This assumption excludes cyclic connections which are not defined in this version of the Flory-Stockmayer theory [63]. There is also a slight difference between the number of binding sites and the number of actin monomers on a filament since binding sites may exist on the interface of two or more actin monomers instead of on a single F-actin monomer. Nevertheless, as the length of an actin filament increases, this difference tends to be negligible. In the chemical kinetic description, we have not included the connections of a single linker or a motor to a single binding site ($F_c \cdot L$ and $F_c \cdot M$). These connections do not alter the connectivity percolation since they do not connect actin monomers to larger networks. We have also simplified the connections made by a brancher to a single binding site ($F_c \cdot B \cdot F_m$). We recognize that it is possible that a brancher may simultaneously connect to multiple F-actin monomers. Nevertheless, the connections between branchers, mother filaments, and daughter filaments do not alter the connectivity percolation since actin monomers attached to branchers are already in the same cluster.

The heterogeneous distribution of binding sites in MEDYAN causes fewer binding sites to be available to be bound by linkers due to the small search distance of the α-actinin linkers ($d_C^{min}$ = 30 nm, $d_C^{max}$ = 40 nm). The difference in the concentration of bound branchers ($[F_c \cdot B \cdot F_m]$) can be explained because of the lower concentration of non-polymerized G-actin in the chemical kinetic model. This lesser concentration of G-actin is due to a faster effective polymerization rate in the chemical kinetic model than that of the MEDYAN model since collisions between actin filaments and the boundary are not represented in the chemical kinetic model. The faster effective polymerization rate leads to an increase of available binding sites ($[F_c]$) in the chemical kinetic model, which in turn allows the branching reaction to occur earlier. Finite-size effects exist in MEDYAN. While such effects can be reproduced in a chemical kinetic model, we consider the chemical kinetic model, which assumes an infinite system, to be sufficient to explain the processes happening during the early stages of the connectivity percolation in the network.

## Conclusion

In this work, we have used a generalization of the Flory-Stockmayer theory of percolation to show three possible phase regimes for a cytoskeletal network depending on the connectivity achieved by motors, linkers and branchers. These connections give rise to local or global contraction depending on the percolation regime. Actin-binding proteins modulate the structure and dynamics of the network, allowing the cell to exhibit different behavior and functions. We also show that Arp2/3 increases the connectivity of the network when the concentration of Arp2/3 is lower than the concentration of F-actin monomers.

## Methods

We have modeled reactions between actin filaments (F-actin), monomeric globular actins (G-actin), and actin-binding proteins using either a mass action chemical kinetics model

described by a system of ordinary differential equations, which we call the chemical kinetic model, or a stochastic mechanochemical model (MEDYAN). We have quantified the number of connections between F-actin monomers in the system and then used a mean field model based on the Flory-Stockmayer theory [49,63,64] to calculate whether the system should behave like a liquid (sol state) or like a semi-solid (gel state).

## Mass action chemical kinetics model

We have modeled five actin binding and unbinding processes in the actomyosin networks using a chemical kinetic model based on mass action kinetics. The equations from the chemical kinetic model parallel the chemical reactions described in MEDYAN [7,39,65–67], a mechanochemical model of actomyosin networks detailed in a section below. The reactions, shown in Table 1, include the polymerization and depolymerization at both the plus ends ($F_p$) and the minus ends ($F_m$) of actin filaments, along with the binding and the unbinding of α-actinin linkers (L), NMIIA motors (M), and the Arp2/3 complex (B) to F-actin. To account for the NMIIA minifilaments, a motor (M) consists of 22.5 myosin molecules, which corresponds to the average number of motor heads in the MEDYAN model [7,21]. The rates for these reactions are shown in Table A in S1 Text. The chemical kinetic model assumes a homogeneous system with perfect mixing, and we do not model the spatial effects of the diffusion of chemical species. The chemical kinetic model also assumes an infinite volume, so the stochasticity of the processes is averaged out.

We have used mass-action kinetics to model the binding of G-actin (G), motors (M), linkers (L), and branchers (B) to actin filaments (F) as well as their unbinding. The kinetic equations replicate the MEDYAN reactions, where a three-body reaction takes place (see section I.A of the S1 Text for more details). We have also included a two-step binding reaction of linkers (L) to actin filaments (F) based on experimental observations (see section I.B of the S1 Text for more details). We have defined three different interfaces for the F-actin monomers: the plus site ($F_p$), the minus site ($F_m$), and the ABP binding site ($F_c$) since actin, motors, linkers, and branchers can be connected to the actin filament monomers through these interfaces as shown in Fig 10.

## Flory-Stockmayer theory

The theory of Flory and Stockmayer [49,64] describes the conversion of monomers first into soluble low molecular weight polymers and then into heavy insoluble gels by branching and crosslinking. In these theories the polymers are described using a mean field model, where the probability of finding a bound functional group depends only on the nature of the functional group. When the fraction of reacted polymers reaches a threshold, then the weighted average monomer size becomes infinite.

**Table 1. Reactions included in the chemical kinetic model.**

| Reaction | Description |
|---|---|
| $F_p + G \underset{k_p^-}{\overset{k_p^+}{\rightleftharpoons}} F_p \cdot F_m + F_p + F_c$ | Actin polymerization at the plus end |
| $F_m + G \underset{k_m^-}{\overset{k_m^+}{\rightleftharpoons}} F_p \cdot F_m + F_m + F_c$ | Actin polymerization at the minus end |
| $F_c + B + G \underset{k_B^-}{\overset{k_B^+}{\rightleftharpoons}} F_c \cdot B \cdot F_m + F_p + F_c$ | Brancher binding |
| $F_c + L + F_c \underset{k_C^-}{\overset{k_C^+}{\rightleftharpoons}} F_c \cdot L \cdot F_c$ | Linker binding |
| $F_c + M + F_c \underset{k_M^-}{\overset{k_M^+}{\rightleftharpoons}} F_c \cdot M \cdot F_c$ | Motor binding |

Tavares et al. recently developed a generalization of the Flory-Stockmayer theory to describe patchy colloids [51,63], which can be understood as polyfunctional branched monomers. We have used this generalization of the Flory-Stockmayer theory to calculate when the percolation transitions occur. The crosslinking probabilities ($\theta_{\alpha \to \beta}$) were calculated using the ratio of the concentration of species in the bound state ($[\alpha \cdot \beta]$) to the total concentration of the species ($[\alpha]_T$) as quantified from the chemical kinetic model or the MEDYAN simulations (Eq 1).

$$\theta_{\alpha \to \beta} = \frac{[\alpha \cdot \beta]}{[\alpha]_T} = \frac{[\alpha \cdot \beta]}{[\alpha] + [\alpha \cdot \beta]} \tag{1}$$

Where $\theta_{\alpha \to \beta}$ is the probability of having an F-actin monomer connected through the site $\alpha$ to the site $\beta$ of another F-actin monomer, and $\alpha$ and $\beta$ can be the plus site (p), the minus site (m), or the actin-binding site (c) (Fig 10).

The probabilities of having an F-actin monomer connected to another F-actin monomer from one site to another are shown in Eq 2.

$$\theta_{p \to m} = \frac{[F_p \cdot F_m]}{[F_p \cdot F_m] + [F_p]}$$

$$\theta_{m \to p} = \frac{[F_p \cdot F_m]}{[F_p \cdot F_m] + [F_c \cdot B \cdot F_m] + [F_m]}$$

$$\theta_{c \to c} = \frac{2[F_c \cdot L \cdot F_c] + 2[F_c \cdot M \cdot F_c]}{2[F_c \cdot L \cdot F_c] + 2[F_c \cdot M \cdot F_c] + [F_c \cdot B \cdot F_m] + [F_c]}$$

$$\theta_{c \to m} = \frac{[F_c \cdot B \cdot F_m]}{2[F_c \cdot L \cdot F_c] + 2[F_c \cdot M \cdot F_c] + [F_c]}$$

$$(2)$$

Where $\theta_{p \to m}$ and $\theta_{m \to p}$ are the probability that an F-actin monomer plus site ($F_p$) is connected to the minus site ($F_m$) of another F-actin monomer and vice versa through actin filament polymerization. The probability that F-actin monomer binding site ($F_c$) is connected to the binding site ($F_c$) of another F-actin monomer is denoted as $\theta_{c \to c}$. Connections through the binding sites are formed by motor and linker binding. Finally, $\theta_{c \to m}$ and $\theta_{m \to c}$ are the probability that F-actin monomer binding site ($F_c$) is connected to the minus site ($F_m$) of another F-actin monomer, and vice versa through brancher binding.

$[F_p \cdot F_m]$ is the concentration of plus sites bound to minus sites, as in polymerized F-actin, $[F_c \cdot L \cdot F_c]$ is the concentration of pairs of F-actin monomers bound through the binding sites with linkers, $[F_c \cdot M \cdot F_c]$ is the concentration of pairs of F-actin monomers bound through the binding sites with motors, and $[F_c \cdot B \cdot F_m]$ is the concentration of F-actin monomers bound with a brancher. $[F_p]$ is the concentration of unbound plus sites of F-actin monomers, $[F_m]$ is the concentration of unbound minus sites of F-actin monomer and $[F_c]$ is the concentration of unbound binding sites. The probability that an F-actin is connected to an infinite cluster is detailed in the section I.C of the S1 Text, the solution for a simple case of linkers and actin is detailed in the section I.D of S1 Text.

## Rigidity percolation

To understand how a rigid lattice is formed in the network we use Maxwell counting, which has also been used as a first step to understand the rigidity of glasses [52,53,62,68–73]. In short this counting procedure is based on the fact that the number of floppy modes per connection (f) is related to the number of degrees of freedom per F-actin monomer (g) minus the number

of constraints given by other connections. (Eq 3)

$$f = g - \sum_{\alpha}^{z} \theta_{\alpha} b = 6 - \frac{1}{2} \sum_{\alpha} \sum_{\beta} b_{\alpha} \to \beta \frac{\theta_{\alpha} \to \beta \left(1 \frac{P_s P_s}{Q_{\alpha} Q_{\beta}}\right)}{1 - P_s} \tag{3}$$

Where p is the probability of forming a contact, z is the coordination number and b is the number of constraints given by the connection. Every F-actin monomer has 6 degrees of freedom in 3D space (g = 6): 3 translational degrees of freedom and 3 rotational degrees of freedom. In addition, the F-actin can connect through 3 possible sites (z = 3). When an F-actin monomer connects to another F-actin monomer, the system loses degrees of freedom depending on the rigidity of the connection, *b*. For this model we considered that each direct connection between two F-actin monomers accounts for a loss of 6 degrees of freedom (b = 6) since we assume each actin filament is a rigid object. For example, when there is no connection between two F-actin monomers there would be two separate filaments with 6 degrees of freedom each (a total of 12 degrees of freedom). When the connection forms, the system contains only one rigid filament with 6 degrees of freedom.

Similarly, rigid connections of two F-actin monomers with linkers, motors, and branchers also account for a loss of 6 degrees of freedom (b = 6). The connectivity percolation is the same as the rigidity percolation (b = g) when the connections are rigid.

On the other hand, when the connections of two F-actin monomers with linkers and motors are floppy (b < 6) the connectivity percolation is not the same as the rigidity percolation, since more than one connection is needed to make the system rigid.

## Coarse-grained mechanochemical model of actomyosin systems (MEDYAN)

We have used an elegant coarse-grained mechanochemical model of actomyosin systems called MEDYAN (Mechanochemical Dynamics of Active Networks) developed by Papoian and his group [7,39,65–67]. MEDYAN models both stochastic chemical reactions and deterministic mechanical representations of far-from-equilibrium systems. In this study, we have included some important actin-binding proteins in actomyosin networks: non-muscle myosin IIA heavy chain (NMIIA) motors, α-actinin linkers, and actin-related protein complex 2/3 (Arp2/3) branchers, all in a fixed geometry (See section I.E of the supplementary information for more details).

## Supporting information

**S1 Text. Model details.** Detailed description of the methods used for the chemical kinetic model, the generalization of the Flory-Stockmayer theory, and the coarse-grained mechanochemical model.
(PDF)

## Acknowledgments

We thank both Neal Waxham and Garegin Papoian for their helpful discussions and acknowledge with gratitude the use of the MEDYAN code provided by Dr. Papoian.

## Author Contributions

**Conceptualization:** Carlos Bueno, James Liman, Peter G. Wolynes.

**Data curation:** Carlos Bueno, James Liman.

**Formal analysis:** Carlos Bueno.

**Funding acquisition:** Margaret S. Cheung, Peter G. Wolynes.

**Investigation:** Carlos Bueno, James Liman, Nicholas P. Schafer.

**Methodology:** Carlos Bueno, James Liman, Nicholas P. Schafer, Peter G. Wolynes.

**Project administration:** Peter G. Wolynes.

**Resources:** Margaret S. Cheung, Peter G. Wolynes.

**Software:** Carlos Bueno, James Liman.

**Supervision:** Nicholas P. Schafer, Margaret S. Cheung, Peter G. Wolynes.

**Validation:** Carlos Bueno, James Liman, Nicholas P. Schafer, Margaret S. Cheung, Peter G. Wolynes.

**Visualization:** Carlos Bueno, James Liman, Peter G. Wolynes.

**Writing – original draft:** Carlos Bueno, James Liman, Margaret S. Cheung, Peter G. Wolynes.

**Writing – review & editing:** Carlos Bueno, James Liman, Nicholas P. Schafer, Margaret S. Cheung, Peter G. Wolynes.

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
