## [Decision Letter · Decision Letter 0]

21 Jan 2022

Dear Prof. Wolynes,

Thank you very much for submitting your manuscript "A generalized Flory-Stockmayer kinetic theory of connectivity percolation and rigidity percolation of cytoskeletal networks" for consideration at PLOS Computational Biology.

As with all papers reviewed by the journal, your manuscript was reviewed by members of the editorial board and by several independent reviewers. In light of the reviews (below this email), we would like to invite the resubmission of a significantly-revised version that takes into account the reviewers' comments.

We cannot make any decision about publication until we have seen the revised manuscript and your response to the reviewers' comments. Your revised manuscript is also likely to be sent to reviewers for further evaluation.

Sincerely,

Nir Gov

Associate Editor

PLOS Computational Biology

Daniel Beard

Deputy Editor

PLOS Computational Biology

Reviewer's Responses to Questions

**Comments to the Authors:**

Reviewer #1: The authors have compared a newly developed kinetic theory for the growth of actomyosin networks with computer simulations in the software environment MEDYAN. The kinetic theory is a new version of the Flory-Stockmayer theory for gelation and similar to the one presented in Ref. 48 by Sciortino and coworkers for patchy particles with three interaction sites (JCP 2010). Here the three different interaction sites for F-actin are minus end, plus end and a lateral binding site for linkers, branchers and motors. The comparison with MEDYAN shows that the kinetic theory works surprisingly well. The big advantage is that it can be interrogated for questions that are not so easily answered in computer simulations and experiments, in particular for the percolation transition by Maxwell counting.

This work is a nice theory extension to the MEDYAN-simulations of branched networks published by the authors as Ref. 19 (PNAS 2020) and certainly deserves publication somewhere. In my view, however, it would be much better suited for more theoretically oriented journals such as JCP (physical chemistry community) or PRE (statistical physics community). For PLOS Comp Biol, I am missing the direct biological relevance. As explained in the following, in my view there are several issues from the biophysics point of view.

In the cell, it just does not happen that all described processes compete with each other at once. Most importantly, myosin needs bipolar actin to contract actin, but branched networks cannot be bipolar. A great experimental demonstration of the importance of bipolar actin for contraction was given by Manuel Thery and Laurent Blanchoin with micropatterning (compare Reymann, Anne-Cécile, et al. "Actin network architecture can determine myosin motor activity." Science 336.6086 (2012): 1310-1314 and Ennomani, Hajer, et al. "Architecture and connectivity govern actin network contractility." Current Biology 26.5 (2016): 616-626). This important subject is not discussed here and a reader not familiar with actomyosin might overlook that branched networks cannot contract.

In general the authors focus most of their discussion on Arp2/3, but as far as I can see, the famous 70 degree angle also implemented in their PNAS-paper with MEDYAN is not used here. It is my understanding that this is be a severe limitation of the Stockmayer-like approach. On the other hand, this is an essential element of the biological system, for example because it has been shown that branched actin networks can undergo phase transitions between different architectures due to this 70 degree branching angle (compare Mueller, Jan, et al. "Load adaptation of lamellipodial actin networks." Cell 171.1 (2017): 188-200).

I am also not sure how realistic is the representation of the motor elements. Single myosin II molecules do not contract actin networks, but myosin II minifilaments do. Minifilaments need bipolar actin and use dozens of motor heads to bind to it. This has been implemented in Cytosim (Cortes, Daniel B., et al. "Bond type and discretization of nonmuscle myosin II are critical for simulated contractile dynamics." Biophysical journal 118.11 (2020): 2703-2717), which is an alternative to MEDYAN, but it is not clear how this is being done here and how well this corresponds to experiments.

In my understanding, the clearest relations to experiment could be the predictions on percolation, but it is not explained if there are experimental data that back these up. Experiments are mentioned in passing in Fig. 4, but only in a negative manner (not relevant).

Very minor comments

Apart from MEDYAN and Cytosim, one should also mention AFINES, as done in the PNAS-paper.

Line 145 typo: In the main the transient concentrations

Line 356 typo: In our previous works [19], we modeled linker binding as a termolecular reaction

Reviewer #2: uploaded

**Have the authors made all data and (if applicable) computational code underlying the findings in their manuscript fully available?**

Reviewer #1: Yes

Reviewer #2: **No: **MEDYAN code was not provided

PLOS authors have the option to publish the peer review history of their article (what does this mean?). If published, this will include your full peer review and any attached files.

Reviewer #1: No

Reviewer #2: No
---

## [Decision Letter · Decision Letter 1]

14 Apr 2022

Dear Prof. Wolynes,

We are pleased to inform you that your manuscript 'A generalized Flory-Stockmayer kinetic theory of connectivity percolation and rigidity percolation of cytoskeletal networks' has been provisionally accepted for publication in PLOS Computational Biology.

Best regards,

Nir Gov

Associate Editor

PLOS Computational Biology

Daniel Beard

Deputy Editor

PLOS Computational Biology

Reviewer's Responses to Questions

**Comments to the Authors:**

Reviewer #1: The authors have answered well to all issues raised by the two reviewers. It is clear that this work has a focus on the simple understanding of the connectivity (percolation) transition in actomyosin networks, and is not geared towards explaining the details of the different actomyosin architectures found in living cells. As it stands, the model compares best to the disordered networks of the lamellum or the cortex, but would not describe the lamellipodium (cannot contract) or stress fibers (bundles, not networks). As I commented earlier, the agreement between the kinetic theory and the computer simulations with MEDYAN is surprisingly good and deserves publication. Although I still think that a physical chemistry or theoretical physics journal would also be a good match, I agree with the authors that PLOS Computational Biology is a good choice to bring these quantitative results to the attention of the biological community. Because it contains a balanced discussion of the relevance for experiments, this work satisfies the criteria for publication in PLOS Computational Biology.

Reviewer #2: The responses of the authors to the various questions raised by myslef and reviewer 1 show that the paper novelty with regard to its applicability to realistic actomyosin systems is moderate. I therefore believe it does not meet the high standards of PLOS COMP BIO. It is highly suitable for PLOS ONE or for a statistical physics or physical chemistry journal in its present form.

**Have the authors made all data and (if applicable) computational code underlying the findings in their manuscript fully available?**

Reviewer #1: Yes

Reviewer #2: Yes

PLOS authors have the option to publish the peer review history of their article (what does this mean?). If published, this will include your full peer review and any attached files.

Reviewer #1: No

Reviewer #2: No

---

## [Editor Report · Acceptance letter]

4 May 2022

PCOMPBIOL-D-21-02017R1 

A generalized Flory-Stockmayer kinetic theory of connectivity percolation and rigidity percolation of cytoskeletal networks

Dear Dr Wolynes,

I am pleased to inform you that your manuscript has been formally accepted for publication in PLOS Computational Biology. Your manuscript is now with our production department and you will be notified of the publication date in due course.

With kind regards,

Livia Horvath
